# Slot Structured World Models

## Abstract

The ability to perceive and reason about individual objects and their interactions is a goal to be achieved for building intelligent artificial systems. State-of-the-art approaches use a feed-forward encoder to extract object embeddings and a latent graph neural network to model the interaction between these object embeddings. However, the feedforward encoder can not extract *object-centric* representations, nor can it disentangle multiple objects with similar appearance. To solve these issues, we introduce *Slot Structured World Models* (SSWM), a class of world models that combines an *object-centric* encoder (based on Slot Attention) with a latent graph-based dynamics model. We evaluate our method in the Spriteworld benchmark with simple rules of physical interaction, where Slot Structured World Models consistently outperform baselines on a range of (multi-step) prediction tasks with action-conditional object interactions. All code to reproduce paper experiments is available from `https://github.com/Anonymous/Slot-Structured-World-Models`.

## 1 Introduction

The ability to distinguish individual components of a visual scene and reason about their interaction is a key aspect of human cognition (Spelke & Kinzler, 2007). It allows us to build a solid comprehension of our environment and is therefore also considered an important requirement for artificially intelligent systems (Battaglia et al., 2018). Ideally, we obtain models that take in raw images, flexibly represent the objects in the scene, and can predict the effect of actions on individual objects and their interactions (Ha & Schmidhuber, 2018; Moerland et al., 2023). Several papers have indeed addressed this challenge (Van Steenkiste et al., 2018; Kipf et al., 2019; Watters et al., 2019b). A particularly successful idea is to extract the objects in the scene and use graph neural networks (GNN) (Wu et al., 2020) to model the pairwise interaction between objects, which achieved state-of-the-art results as 'Contrastively Learned Structured World Models' (C-SWM) (Kipf et al., 2019).

The current GNN-based approach does however have remaining challenges, as noted by the authors of C-SWM as well (Kipf et al., 2019). The method uses a feedforward encoder to embed the scene into a fixed set of embeddings for the latent GNN model, an approach that has several limitations. First of all, the fixed feedforward encoder cannot disambiguate multiple objects with the (approximate) same appearance: they will be detected in the same feature map (see Fig. 1 for an illustration of this problem). Moreover, the number of discoverable objects is fixed in the architecture and cannot therefore vary at inference time.

To overcome the above limitations, we propose to instead feed the latent GNN-based transition model with *an object-centric encoder* (Engelcke et al., 2019; Greff et al., 2019; Burgess et al., 2019; Locatello et al., 2020; Biza et al., 2023). Such encoders learn to return a set of embeddings that each represent information about an individual object in the scene. One successful approach is Slot Attention (SA) (Locatello et al., 2020), which utilizes an (iterative) competitive attention mechanism between the object-specific slots to force individual objects into distinct slots. These methods repeatedly apply the same encoder to extract each object (sharing information), can disentangle similar objects due to the competitive attention, and can adjust the number of initialized slots at inference time. As such, object-centric encoders such as Slot Attention provide the exact characteristics we desire for downstream use in GNN-based dynamics models.

This paper therefore proposes a new type of dynamics model that embeds *an object-centric encoder and a GNN-based world model*. In particular, we use Slot Attention to generate embeddings for a latent GNN

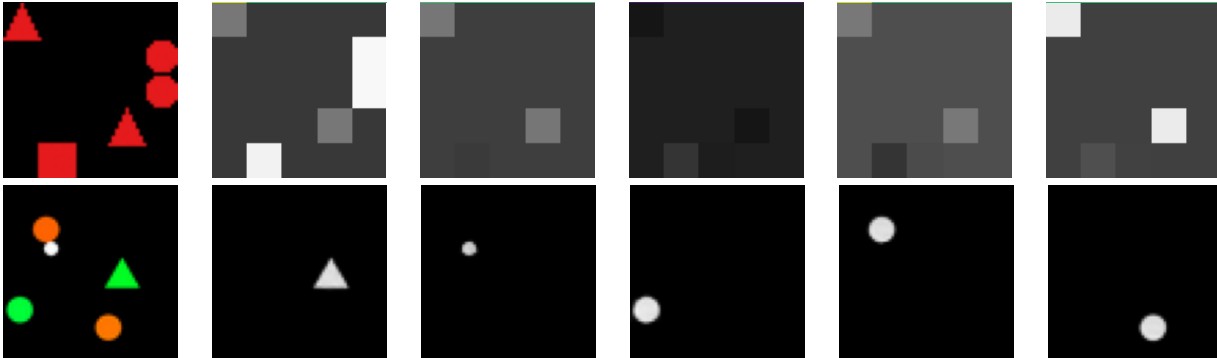

Figure 1: Comparison of object masks learned by C-SWM (baseline, top row) (Kipf et al., 2019) on Shapes 2D and SSWM (our method, bottom row) on *Interactive Spriteworld*. The left image of each row shows the input image, while the five images next to it show the learned object masks (encoder output) that enter the GNN. **Top**: The input image contains two duplicate objects (red circles and red triangles), which the C-SWM encoder cannot disentangle and instead conflates in a single embedding (for example, the two red circles end up in both the first and fourth embedding). These embeddings will make modeling of pairwise object interactions in the downstream GNN less effective. **Bottom**: In contrast, our SSWM method can disentangle the duplicate items in the input image (orange circles) into distinct slots, due to the object-centric competitive attention mechanism in the Slot Attention encoder. This allows for more effective modeling of object interaction in the downstream GNN, as we show in the Results section and in B. We made the comparison on different environments as C-SWM could not learn any useful representation in Interactive Spriteworld. This does not affect the significance of the figure as its purpose is to exhibit the behaviors of the two methods when dealing with visually identical objects.

dynamics model (inspired by C-SWM), which we call *Slot Structured World Models* (SSWM). The high-level architecture of SSWM is shown in Fig. 2. However, note that the idea to combine object-centric encoders and GNN-based dynamics models is more general, and one can easily swap Slot Attention for any other object-centric embedding method (as long as the method produces a set of feature vectors that bind individual objects).

To test our approach we extend the well-known Spriteworld benchmark (Watters et al., 2019a), introduced for object-centric learning in COBRA (Watters et al., 2019b), with physical rules of interaction. While in the original Spriteworld an object pushed into another object would start to overlap, in our new *Interactive Spriteworld* environment the second object will be pushed away. This ensures objects actually interact with each other, and also demands richer representations (since the exact shape of an object starts to matter, while in the previous setting their location and velocity would suffice). Experimental evaluation on a range of Interactive Spriteworld tasks shows that SSWM consistently outperforms the state-of-the-art baseline C-SWM in 1-step, 5-step, and 10-step predictive accuracy. Quantitative evaluation indicates that SSWM indeed makes accurate predictions, although slight deviations do start to accumulate over longer prediction horizons.

In summary, this paper proposes SSWM, the first learned dynamics model that:

- can isolate individual objects *and* reason about their (action-conditional) interactions from raw pixel input

- can disambiguate between multiple objects with similar appearance

- numerically outperforms the state-of-the-art object-centric dynamics model C-SWM on (multi-step) prediction tasks

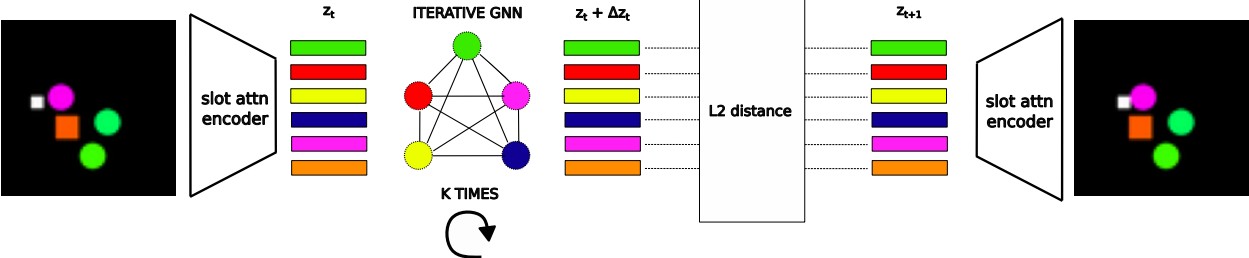

Figure 2: Architectural design of Slot Structured World Models. Given an input image (left), we use a pretrained Slot Attention encoder to produce a set of object-centric embeddings ($z_t$) that capture the objects (and background) in the scene. These latent embeddings are then fed together with the chosen action into an iterative GNN transition module (defined in Algorithm 1) to predict the change in the next latent state ($\Delta z_t$). The graph-based transition model is trained to minimize the (object-wise) L2-norm between the latent prediction ($z_t + \Delta z_t$) and the Slot Attention embedding of the true next state ($z_{t+1}$). The slot attention encoder used to obtain the targets for our loss is the same exact encoder we use to produce the state embeddings.

## 2  Background

The architecture presented by (Kipf et al., 2019) in C-SWM, is structured in three different sub-modules: object extractor, object encoder, and relational transition model. The object extractor is a function $E_{ext}(s_t) = \{m_1, ..., m_k\}$ mapping an RGB image $s_t$ into a set of $K$ feature maps meant to represent a mask binding to one of the $K$ objects ($K$ is fixed). It is implemented as a fully convolutional network whose last layer produces at least $K$ feature maps (at least one per object) and is activated by sigmoid (to obtain feature maps interpretable as masks). The object encoder shares parameters across the objects and is a function $E_{enc}(m_i) = z_i$ embedding each of the $K$ masks to a latent vector $z_i \in \mathbb{R}^{\mathcal{D}}$. The relational transition model is a function $T(z_t^k, a_t^k) = \Delta z_t^k$ computing the update for each of the abstract representation $z_t^k$ given an action $a_t^k$ associated. It is implemented as a Graph Neural Network (GNN) (Scarselli et al., 2008) using the following message-passing updates:

$$e_t^{(i,j)} = f_{edge}([z_t^i, z_t^j]) \tag{1}$$

where $e_t^{(i,j)}$ is the edge (the relationship) between the nodes $z_t^i$ and $z_t^j$, and $f_{edge}([z_t^i, z_t^j])$ is a simple MLP taking as input the concatenation of the vectors $z_t^i$ and $z_t^j$.

$$\Delta z_t^j = f_{node}([z_t^j, a_t^j, \sum_{i \neq j} e_t^{(i,j)}]) \tag{2}$$

where $\Delta z_t^j$ is the update of the object $z_t^j$ given the application of action $a_t^j$ and the sum of its incoming edges. The entire architecture is trained to minimize the loss function:

$$\mathcal{L} = H + max(0, \gamma - \tilde{H}) \tag{3}$$

$$H = \frac{1}{K} \sum_{k=1}^{K} d(z_t^k + \Delta z_t^k, z_{t+1}^k), \quad \tilde{H} = \frac{1}{K} \sum_{k=1}^{K} d(\tilde{z}_t^k, z_{t+1}^k) \tag{4}$$

with $H$ being the average Euclidean distance between the predicted next states (factorized) $z_t^k + \Delta z_t^k$ and the encoding of the real next states $z_{t+1}^k$. This part of the objective is used to optimize the performance

of the transition model. $\tilde{H}$ is the average Euclidean distance between the encoding of a state $\tilde{s}_t$ randomly sampled from the experience replay. This part of the objective is meant to prevent the encoder from similarly representing different states. In other words, the first part of the objective moves the target towards positive examples while the second distances it from negative ones.

## 3 Related Work

**Object-centric representation learning**   Several lines of work in this field proposed successful unsupervised approaches in tasks such as object discovery. Some examples of these approaches are Engelcke et al. (2019); Greff et al. (2019); Burgess et al. (2019); Locatello et al. (2020), which learn to represent raw images as a set of latent vectors binding to individual objects. Different from the other works mentioned (that use multiple encode-decode steps), Slot Attention (Locatello et al., 2020) manages to perform just one encoding step thanks to a simple yet effective iterative attention module. This characteristic makes Slot Attention much more computationally efficient than its precursors. For this and other reasons (such as data and training efficiency), we chose this model for the scope of our work. For the same reasons, this approach has been further extended by more recent works. In particular, works such as Chang et al. (2022); Jia et al. (2022) propose extensions to improve the slot optimization process, while the more recent Invariant Slot Attention (Biza et al., 2023) introduces a method to make Slot Attention's representations invariant to features such as position, scale, and rotation.

**Object-based Models of Environments**   Given the structured nature of many environments where multiple entities (agents and objects) interact, learning a robust and accurate model of such environments requires specific architectural biases. In line with this premise, numerous previous approaches such as (Sukhbaatar et al., 2016; Chang et al., 2016; Battaglia et al., 2016; Watters et al., 2017; Hoshen, 2017; Wang et al., 2018; Van Steenkiste et al., 2018; Kipf et al., 2018; Sanchez-Gonzalez et al., 2018; Xu et al., 2019; Kipf et al., 2019) adopted solutions based on graph neural networks to build structured world models. Among these methods, C-SWM stands out for the ability to learn states' representations and a transition model simultaneously and without relying on any pixel-based loss. Yet this method presents an encoder that cannot separate the information of objects with identical appearances. Another relevant approach in this field is COBRA (Watters et al., 2019b), which, similarly to our approach, adopts a pretrained object-centric model (MONet (Burgess et al., 2019)) to obtain structured representations. Unlike C-SWM and this work, COBRA does not model relationships between objects, as its transition model is not implemented as a graph neural network.

## 4 Environment

The Spriteworld environment (Watters et al., 2019a), introduced in COBRA (Watters et al., 2019b), is a visual benchmark task where objects of varying shapes and colors can be moved around. However, when two of these objects are pushed against each other, they start to overlap (instead of pushing against each other). This lack of physical object interaction makes that the dynamics of Spriteworld can be encoded by just the object positions (Watters et al., 2019b). To challenge our models in a more complex environment that requires richer object representations, we therefore extend Spriteworld with simple rules of physics: *Interactive Spriteworld* (available at `https://github.com/Anonymous/Interactive-Spriteworld`), in which an embodied agent can move objects around (Fig. 3).

The environment consists of a square arena with a black background and five objects. At the beginning of every episode, the shape of each object is sampled from a discrete and uniform distribution of three shapes (circle, square, and triangle), the position is sampled from a continuous distribution covering the entire space, and the (plain) color is sampled from a continuous distribution along three channels (HSV). The agent is represented by a white sprite of smaller size. This agent can take one of four possible actions (move down, up, left, or right) and moves other objects by colliding with them.

The first row of Figure 3 shows some example observations from Interactive Spriteworld. We can clearly observe scenes with objects of similar or identical appearance, which we expect the C-SWM encoder to

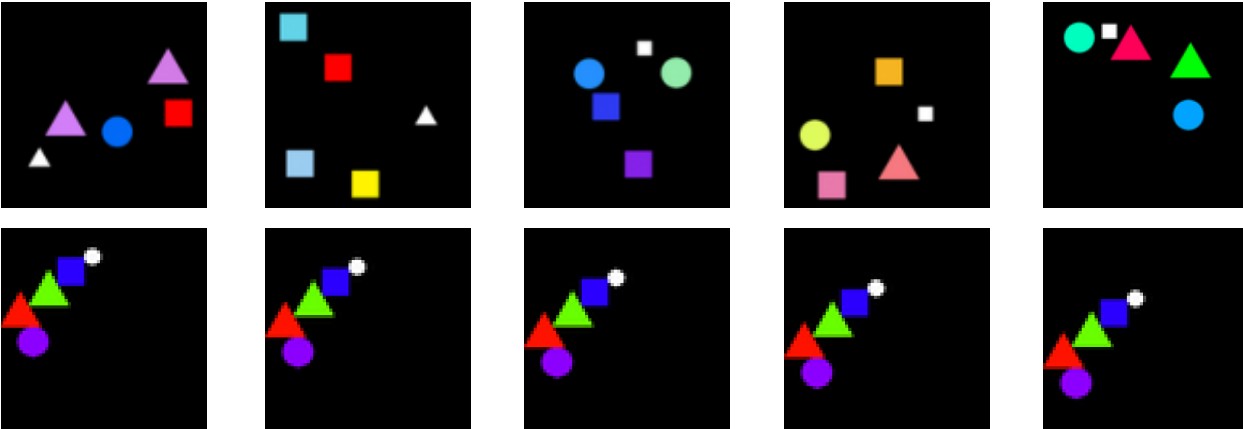

Figure 3: Example states from Interactive Spriteworld. The top row shows five samples from different episodes. The bottom row shows five consecutive timesteps of a particular episode where the agent (white circle) keeps moving down, carrying all objects in the scene down as well (since they push against each other).

struggle with (see Fig. 1). The second row illustrates the implemented simple rules of physics: if a moving sprite/object A hits a static object B, B acquires A's motion, and so on until the chain stops. To model these dynamics, a learned transition model needs representations containing information about each object's position, shape, and size, to determine the exact moment of contact between two objects.

## 5 Methodology

The overall architecture of Slot Structured World Models is shown in Fig. 2. We first encode the input image (left part of the figure) with the object-centric Slot Attention encoder (Locatello et al., 2020). This Slot Attention encoder is pretrained separately, with full details provided in Appendix A.1. The output of the encoder is a set of object-centric representations ($z_t = \{z_t^1, .., z_t^K\}$), where $K$ denotes the number of slots. These slot vectors are then fed into the graph neural network to model the interaction between these objects (left-middle part of the figure). The output of the GNN is a set of object-centric delta vectors ($\Delta z_t$) that predict the change in each object's latent representation given the chosen action. We train the GNN on an object-wise L2 loss (right-middle of the figure) between the predicted next states ($z_t + \Delta z_t$) and the encoding of the true next states ($z_{t+1}$) (right-part of the figure), similar to the first term of the loss in Eq. 3. We drop the second term of the contrastive loss since the pretrained SA encoder already provides us with a well-structured latent representation space. Note that our Slot Attention encoder is pretrained, but the whole architecture could also be trained on the same loss in an end-to-end fashion.

**SA design**  Since the loss of the transition model compares the predicted slot vectors with the ones resulting from the encoding of the next state in a pair-wise fashion, we prefer to maintain their ordering constant. This prevents computationally expensive sorting based on similarity metrics. For this reason, we learn fixed slot initializations instead of sampling them from a (learned) normal distribution. However, note that this choice does fix the number of slots at inference time in our experiments.

**GNN design**  We need to further adapted the C-SWM transition model to be suitable for a more complex environment such as *Interactive Spriteworld*. As mentioned in Section 4, colliding objects may cause a chain of movement in several objects. For this reason, a single round of message passing is not sufficient to model the dynamics of the environment, and we need to iterate this process multiple times. To allow the GNN to take into account previous node updates, the edge function in Equation 1 is turned into $f_{edge}([z_{t+1}^i, z_{t+1}^j, \Delta z_t^i])$. The intuition is that the update $\Delta z_t^i$ contains information about the direction of the transition of $z_t^i$ into $z_{t+1}^i$. Therefore, in case of collision between $z_{t+1}^i$ and $z_{t+1}^j$, the edge function can predict a relation vector

---

**Algorithm 1** Iterative GNN Module

---

**Require:** $\mathcal{S} = \{z^1, ..., z^K\} \in \mathbb{R}^{K \times D}$ Slots vectors, $a_t$ action

$\quad z_0^i \leftarrow \{0\}^D$
$\quad i, j \leftarrow get\_inds\_from\_adj\_matrix(K)$
$\quad e_0^{(i,j)} \leftarrow f_{edge}([z_0^i, z_0^j, \Delta z_0^i])$
$\quad \Delta z_1^j \leftarrow f_{node}([z_0^j, a_t, \sum_{i \neq j} e_0^{(i,j)}])$
$\quad z_1^i \leftarrow z_0^i + \Delta z_1^i$
$\quad$**for** $n = 1 \rightarrow K$ **do**
$\quad\quad e_n^{(i,j)} \leftarrow f_{edge}([z_n^i, z_n^j, \Delta z_n^i])$
$\quad\quad \Delta z_{n+1}^j \leftarrow f_{node}([z_n^j, \{0\}^D, \sum_{i \neq j} e_n^{(i,j)}])$
$\quad\quad z_{n+1}^i \leftarrow z_n^i + \Delta z_{n+1}^i$
$\quad$**end for**

---

$e^{(i,j)}$ that can be interpreted as the force applied by $z_{t+1}^i$ on $z_{t+1}^j$ given $\Delta z_t^i$. Finally, the node function then predicts a node update based on the associated action and the sum of all its incoming edges.

**Iterative GNN procedure** The full iterative GNN procedure of SSWM is shown in Algorithm 1. The module takes as input a set of $K$ vectors $z_t^i \in \mathbb{R}^D$ and an action $a_t$ (one-hot vector). Before starting the loop, the algorithm builds a $K \times K$ adjacency matrix with the diagonal set to zero (to avoid relations between an object and itself) representing a fully connected graph, and initializes the per-object updates $\Delta z_0^i$ to zero (since no object has moved yet). At this point, the edge function is fed with a set of $K \times (K-1)$ vectors obtained as the concatenation of $z_0^i$, $z_0^j$, and $\Delta z_0^i$ with the auxiliary of the adjacency matrix. Then the node updates $\Delta z_1^j$ are obtained as in Equation 2 and the nodes $z_1^i$ are obtained by summing all the nodes $z_0^i$ with the corresponding updates $\Delta z_1^i$. This first phase before the loop is meant to predict the position of the agent after it takes action $a_t$, while the iterative part is meant to predict the transition of every other object based on the forces they apply to each other. For this reason, the actions to be fed to the node function are set to zero.

At every iteration $n > 0$ the edges are computed as $e_t^{(i,j)} = f_{edge}([z_n^i, z_n^j, \Delta z_n^i])$, the node transitions as $\Delta z_{n+1}^j = f_{node}([z_n^j, 0, \sum_{i \neq j} e_t^{(i,j)}])$, and the nodes as $z_{n+1}^i = z_n^i + \Delta z_{n+1}^i$. Finally, since in the worst case, the agent motion causes movement of all the other objects, the number of iterations is set to $K - 1$ where $K$ is the number of discoverable objects.

# 6 Experiments

## 6.1 Metrics

We adopt the same metric as used by Kipf et al. (2019), namely: Hits at rank k (H@k) and Mean Reciprocal Rank (MMR). These metrics allow direct evaluation in latent space instead of having to train a separate decoder.

- **Hits at rank k:** We first rank each predicted object vector based on its distance with the full set of ground-truth encoded object vectors in the entire dataset (similar to Kipf et al. (2019)). The H@k score per object vector is 1 when the rank of the inferred vector does not exceed k, and 0 otherwise. We report the percentage of 'Hits at rank 1' over the entire dataset.

- **Mean Reciprocal Rank:** The MRR is an aggregate of the above ranking, defined as the average of the inverse rank of all the $n$ samples present in the evaluation dataset: MRR $= \frac{1}{N} \sum_{n=1}^{N} \frac{1}{\text{rank}_n}$, where $\text{rank}_n$ is the rank of the $n$-th sample.

Note that latent space evaluation metrics do have some edge cases, about which we further report in Appendix A.2.

Table 1: Quantitative performance of SSWM and C-SWM on Interactive Spriteworld. Results are reported over three prediction horizons (1, 5, and 10 steps) and over three test sets (collision-free, single collision, chain of collisions). The best scores are highlighted in bold. We observe that SSWM outperforms C-SWM on all test scenarios and prediction horizons, with a difference that becomes more pronounced over longer horizons. Standard deviations over four runs are added in grey.

| | Model | 1 Step | | 5 Steps | | 10 Steps | |
|---|---|---|---|---|---|---|---|
| | | H@1 | MRR | H@1 | MRR | H@1 | MRR |
| test 1 | **SSWM** | $\mathbf{98.2}_{\pm 1.3}$ | $\mathbf{98.8}_{\pm 0.8}$ | $\mathbf{93.5}_{\pm 1.1}$ | $\mathbf{95.3}_{\pm 0.8}$ | $\mathbf{88.0}_{\pm 2.1}$ | $\mathbf{92.4}_{\pm 1.1}$ |
| | C-SWM | $72.0_{\pm 18.5}$ | $82.2_{\pm 11.9}$ | $36.0_{\pm 23.4}$ | $53.5_{\pm 19.5}$ | $23.7_{\pm 18.6}$ | $42.7_{\pm 17.7}$ |
| test 2 | **SSWM** | $\mathbf{97.5}_{\pm 1.1}$ | $\mathbf{97.9}_{\pm 0.9}$ | $\mathbf{86.8}_{\pm 0.8}$ | $\mathbf{90.2}_{\pm 1.0}$ | $\mathbf{73.8}_{\pm 1.6}$ | $\mathbf{81.2}_{\pm 0.5}$ |
| | C-SWM | $89.7_{\pm 7.4}$ | $94.2_{\pm 4.1}$ | $50.0_{\pm 14.9}$ | $69.6_{\pm 10.0}$ | $20.3_{\pm 13.2}$ | $48.0_{\pm 10.0}$ |
| test 3 | **SSWM** | $\mathbf{97.2}_{\pm 0.4}$ | $\mathbf{98.0}_{\pm 0.3}$ | $\mathbf{89.0}_{\pm 0.6}$ | $\mathbf{93.5}_{\pm 0.7}$ | $\mathbf{68.3}_{\pm 3.5}$ | $\mathbf{79.6}_{\pm 2.0}$ |
| | C-SWM | $86.3_{\pm 9.0}$ | $92.3_{\pm 5.2}$ | $59.3_{\pm 10.5}$ | $75.5_{\pm 7.5}$ | $19.3_{\pm 12.6}$ | $48.2_{\pm 10.5}$ |
| train | **SSWM** | $\mathbf{100}_{\pm 0.0}$ | $\mathbf{100}_{\pm 0.0}$ | $99.0_{\pm 0.2}$ | $99.1_{\pm 0.2}$ | $95.7_{\pm 0.7}$ | $96.4_{\pm 0.5}$ |
| | C-SWM | $\mathbf{100}_{\pm 0.0}$ | $\mathbf{100}_{\pm 0.0}$ | $\mathbf{100}_{\pm 0.0}$ | $\mathbf{100}_{\pm 0.0}$ | $\mathbf{100}_{\pm 0.0}$ | $\mathbf{100}_{\pm 0.0}$ |

## 6.2 Experimental Setup

All results are averaged over four independent repetitions, with hyperparameters reported in Appendix Table 5. As a baseline we use C-SWM with default feedforward encoder, augmented with the iterative GNN module described Algorithm 1. We further augment the number of feature maps from one to four per object to model the positional information (as in the original paper) as other relevant information. The training dataset consists of $6 \cdot 10^4$ samples, one-third of which were collected with a random policy acting in the environment, and the remaining part from human demonstration to collect a diverse set of complex interactions (which does not occur with random action selection). The test set is divided into three disjointed subsets sorted in ascending order of difficulty, each containing $10^3$ unseen samples structured in 10 episodes of 100 timesteps. The first set contains 10 "collision-free" trajectories where the agent is the only sprite moving, the second set includes several sequences of steps (per episode) where the agent carries one sprite at once, and the last contains complex trajectories where the agent can carry multiple sprites at once or in a chain. This division is meant to highlight the strengths and limits of both classes of models and help to individuate and interpret their behaviors under different settings.

## 6.3 Quantitative Analysis

Table 1 shows quantitative performance measures for both SSWM and C-SWM on the three types of prediction tasks in Interactive Spriteworld, split up into 1-step, 5-step, and 10-step prediction horizons. We first of all observe an increase in performance of SSWM over C-SWM for all studied settings. The first column shows the 1-step prediction accuracy, which achieves a score close to 100% on unseen instances of *Interactive Spriteworld*. Interestingly, the accuracy in scenarios where the action only affects the agent (Test 1) is almost equal to the accuracy in scenarios where the agent moves one object (Test 2) or multiple objects (Test 3). This suggests SSWM is able to accurately generalize over object interactions.

When we increase the prediction horizon to 5 (second column) and 10 (third column) steps, we see that the prediction accuracy starts to decrease for both SSWM and C-SWM. This phenomenon is probably due to *compounding errors*, a well-known phenomenon when unrolling multi-step prediction models (Talvitie, 2014). We also see that SSWM starts to more strongly outperform C-SWM, especially on the longest prediction horizon. This indicates SSWM has better generalization performance about object interactions. Interestingly, C-SWM did manage to minimize the loss function on the training data (bottom row), but did so without encoding indispensable information such as the object shape (a topic to which we return in the next section). This should not be surprising as the contrastive objective of C-SWM does not explicitly

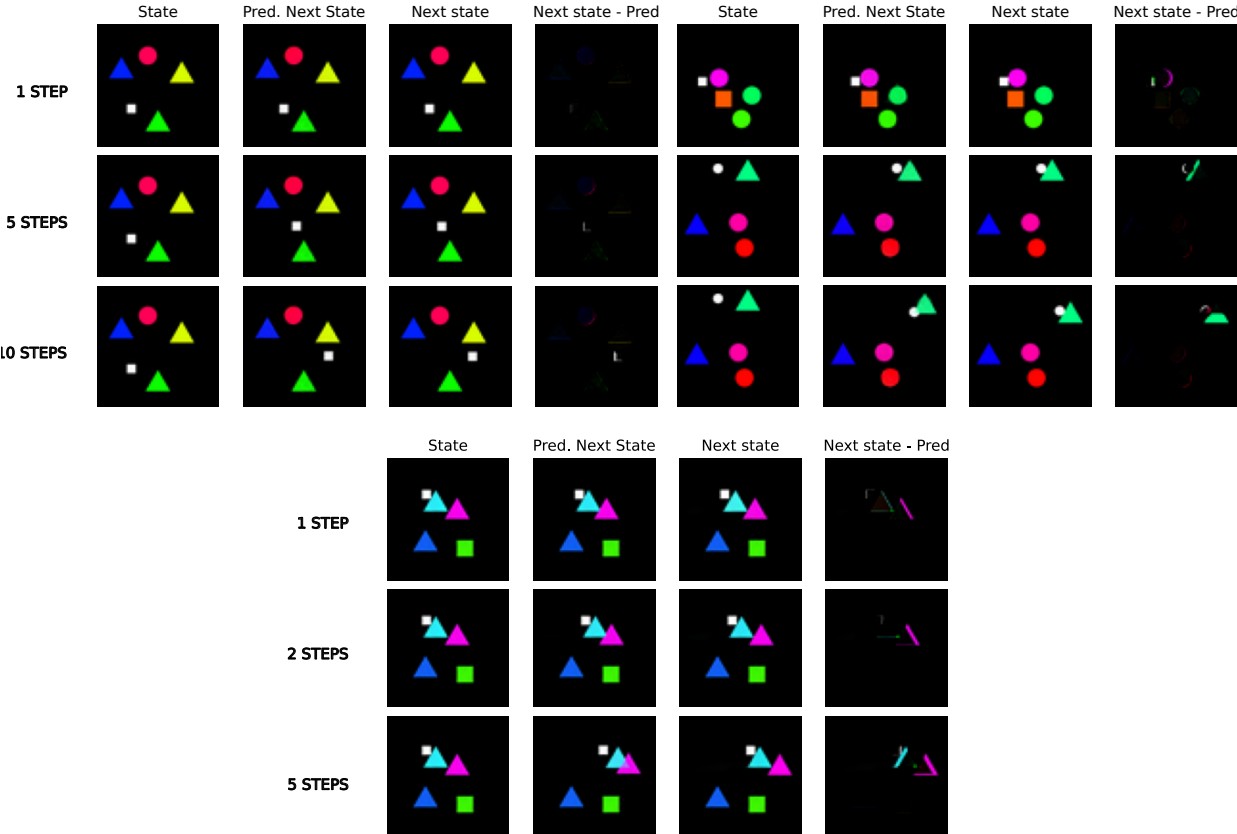

Figure 4: Pixel space decoding of the learned latent space predictions of SSWM. The top-left 3x4 images show predictions when only the agent/sprite is moving, the top-right 3x4 images show predictions when the agent carries one object, and the bottom 3x4 images show predictions when the agent moves multiple objects (that push against each other). For each 3x4 block, the three rows show 1, 5, and 10-step predictions, respectively. The four images in each of these rows show the true source state ('State'), the decoded state predicted by SSWM ('Pred. Next State'), the true state reached after taking the actions ('Next State'), and the prediction error between the latter two ('Next state - Pred'), where full black indicates no error.

lead the encoder in that direction, and the network can find many solutions to uniquely identify objects. In contrast, SSWM appears to better encode all object properties, and as such is able to achieve more accurate long-range predictions.

## 6.4 Qualitative Analysis

We also want to qualitatively evaluate the predictions of our models, ideally in the original pixel space (for interpretability). Therefore, we use the decoder obtained during the Slot Attention pretraining phase to show the predictions SSWM makes. For these visualizations, we use the best model obtained out of the four repetitions. Do note that the dynamics model was never explicitly trained for these pixel space reconstructions, but only received a loss in latent space. For C-SWM, we, of course, do not have this decoder, and instead show the masks produced by the object extractor to evaluate the quality of the embeddings.

**SSWM predictions** Fig. 4 shows examples of predictions made by SSWM in the three test scenarios: only the agent moves (top-left 3x4 block), the agent moves a single object (top-right 3x4 block) or the agent moves multiple objects (bottom 3x4 block). In each block, the three rows show an example of a 1-step (top

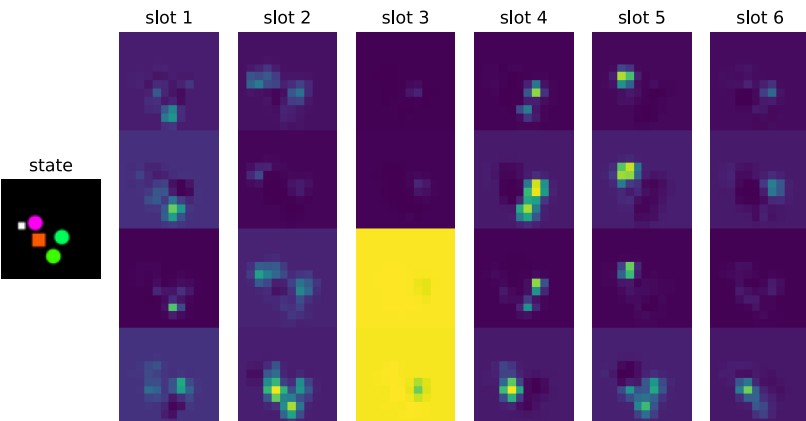

Figure 5: Visualization of the object masks produced by the C-SWM object extractor, for the input images labeled as 'State' (left). We have six slots that each contain four feature maps, which we display above each other. Ideally, each slot uniquely identifies an object and represents its shape, which we need in the downstream prediction task. Instead, we see the model cannot detect exact shapes, nor can it isolate objects per slot (in contrast to the Slot Attention encoder of SSWM, as shown in Fig. 1). This suggests C-SWM learned a solution that does optimize the contrastive objective of 3, but does not encode all relevant information.

row), 5-step (middle row), and 10-step (bottom row) prediction. The four images next to each other for each setting display the start state, the predicted next state, the true next state, and the error between both.

When only the agent moves (top left block of the figure), we see the model predictions are accurate, with a small difference between the predicted and observed next state. In accordance with the quantitative results of the previous paragraph, the prediction does become less accurate as the number of steps increases (due to accumulating errors). The top-right block shows cases in which the agent carriers a sprite in its movement. Again, we see how the model accurately captures the desired movement of both the agent and the object it touches, but deviations accumulate over time. For the 10-step prediction, the model has slightly mispredicted the point of contact between the agent and object (green triangle), and therefore the object ends up too high in the prediction.

The bottom block of the figure shows a scenario where the agent moves multiple objects, pushing against the blue triangle which itself pushes the purple triangle. We can clearly see that SSWM can model such multi-object interactions, since the purple triangle is indeed predicted to move in a coherent fashion with the chosen action. We do again see some slight prediction error, in accordance with earlier observations, that accumulates over time. In agreement with the quantitative results, the compounding error increases faster when multiple objects interact (mostly since more objects can be mispredicted).

**C-SWM representations** We hypothesized that the lack of generalization exhibited by the baseline (C-SWM) is due to the poor quality of the learned embeddings. To investigate this hypothesis, Fig. 7 shows the feature maps obtained per slot by the C-SWM encoder output. Note that each slot consists of four feature maps, which we plot above each other. Clearly, the Interactive Spriteworld tasks require the encoding to represent the exact shape of an object (to determine the point of contact). However, from the visualization, we can clearly see that the C-SWM representations do not represent their exact shape, nor do they even represent them individually. This suggests the CNN-object extractor of C-SWM did not learn filters that isolate objects and contain the necessary features for this task. This observation, together with the quantitative results reported in Table 1, confirms the hypothesis that C-SWM learned a solution that does optimize the objective in Eq. 3 without encoding all relevant information. In contrast, the features learned by the Slot Attention encoder of SSWM do isolate objects and capture their exact shape, as was already shown in the bottom row of Fig. 1. This likely explains the better quantitative performance of SSWM visible in Table 1.

# 7 Conclusion

This paper introduced *Slot Structured World Models*, a simple and flexible framework that combines an object-centric encoder with a GNN-based latent transition model. The full SSWM architecture outperforms the state-of-the-art object-centric dynamics model C-SWM by learning more informative representations, while it is also able to disambiguate multiple objects with similar appearances in a scene. Qualitative analysis indicates that C-SWM overfits the training data without learning meaningful latent representations, whereas SSWM learns better representations through Slot Attention which strongly improves prediction performance.

There are several directions for future work. First, we currently learn a fixed number of slot initializations, since this allows us to directly construct the pairwise latent loss. However, object-centric encoders such as Slot Attention naturally allow a change in the number of latent slots (discoverable objects), by training a slot initialization distribution. We would then of course need to construct the pairwise latent object loss based on similarity metrics, which might make optimization more unstable. Another solution would be to train the whole architecture in an end-to-end fashion. This might also improve the qualitative pixel space predictions: we currently fully rely on the representations learned by SA auto-encoding, but a dynamics loss on the change between frames might put more emphasis on agent and object behavior. A third direction would be to improve multi-step prediction performance, for which we could for example train on a multi-step objective (Abbeel & Ng, 2004). Finally, it would also be interesting to test these models in downstream decision-making tasks, i.e., model-based reinforcement learning.

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

# A    Appendix

## A.1    Slot Attention

The Slot Attention module proposed in Slot Attention Locatello et al. (2020) takes as input a set $\mathcal{X} \in \mathbb{R}^{\mathcal{N} \times \mathcal{D}_{input}}$ of $\mathcal{N}$ feature vectors augmented with positional embeddings, and produces a set $\tilde{\mathcal{S}} \in \mathbb{R}^{\mathcal{K} \times \mathcal{D}}$ of $\mathcal{K}$ slots. The feature vectors are projected in a $\mathcal{D}$-dimensional space through learned linear transformations $k$ and $v$ (respectively keys and values), while the slots (queries) are independently sampled from a Normal distribution defined by learnable parameters $\mu, \sigma \in \mathbb{R}^D$. The slots are then refined through an iterative mechanism involving multiple steps per iteration.

The first step consists of computing the attention matrix

$$\mathcal{A} = Softmax(\frac{k(x) \cdot q(s)^T}{\sqrt{\mathcal{D}_{slot}}}) \tag{5}$$

where $x$ are the input feature vectors and $s$ the sampled slots. The dot product between the input features and the slots relates each slot with parts of the image, then the resultant coefficients are normalized by softmax over the queries to ensure competition between the slots for explaining part of the image. At this point, the updates are obtained by combining the input values with the attention matrix normalized over the slots:

$$\mathcal{U} = W^T \cdot v(x), \quad W_{i,j} = \frac{\mathcal{A}_{i,j}}{\sum_{l=1}^{\mathcal{N}} \mathcal{A}_{l,j}} \tag{6}$$

The final updates of each iteration are obtained by a passage of Gated-recurrent-Unit GRU followed by an MLP with a residual connection.

In order to perform object discovery, the slots are decoded into images and masks that are then combined and summed up to obtain a single image. The training objective is therefore to minimize the mean squared error between the input image and the reconstructed one.

## A.2    Additional Experiments

This section shows some additional experiments initially meant to solve the multi-step prediction issue. Hence, in the first instance, we attributed the difficulty of SSWM in making predictions over longer horizons to Slot Attention's embeddings. We thought that having representations where all the features are entangled (as those of Slot Attention) could be a limit for the latent transition model accuracy (and generalization). Even though disentanglement is certainly a desired property, the next experiments highlighted that the causes of the multi-step prediction issue reside somewhere else. In addition, the next experiments show some limit cases where the metrics involved in our experiments can lead to improper interpretations.

### A.2.1    SSWM

The results shown in this subsection are obtained by replacing Slot Attention with DISA (**?**) which ensures disentangling features such as position, scale, shape, and texture.

The results observable in Table 2 may create the illusion that the disentangled representations provided by DISA suffice to solve the multi-step prediction problem. Unfortunately, Figure A.2.1 shows some visual examples highlighting a huge divergence with the quantitative measures. We care to explain that we observed

Table 2: Results obtained by SSWM by replacing Slot Attention with DISA.

| | 1 Step | | 5 Steps | | 10 Steps | |
|---|---|---|---|---|---|---|
| Test Split | H@1 | MRR | H@1 | MRR | H@1 | MRR |
| test 1 | $100_{\pm 0.0}$ | $100_{\pm 0.0}$ | $90.0_{\pm 0.0}$ | $95.0_{\pm 0.0}$ | $100_{\pm 0.0}$ | $100_{\pm 0.0}$ |
| test 2 | $100_{\pm 0.0}$ | $100_{\pm 0.0}$ | $100_{\pm 0.0}$ | $100_{\pm 0.0}$ | $100_{\pm 0.0}$ | $100_{\pm 0.0}$ |
| test 3 | $100_{\pm 0.0}$ | $100_{\pm 0.0}$ | $100_{\pm 0.0}$ | $100_{\pm 0.0}$ | $100_{\pm 0.0}$ | $100_{\pm 0.0}$ |

also very good qualitative examples using this Encoder, however, we preferred showing our experiments with slot attention as the quantitative results are more in accordance with what we observe in pixel space.

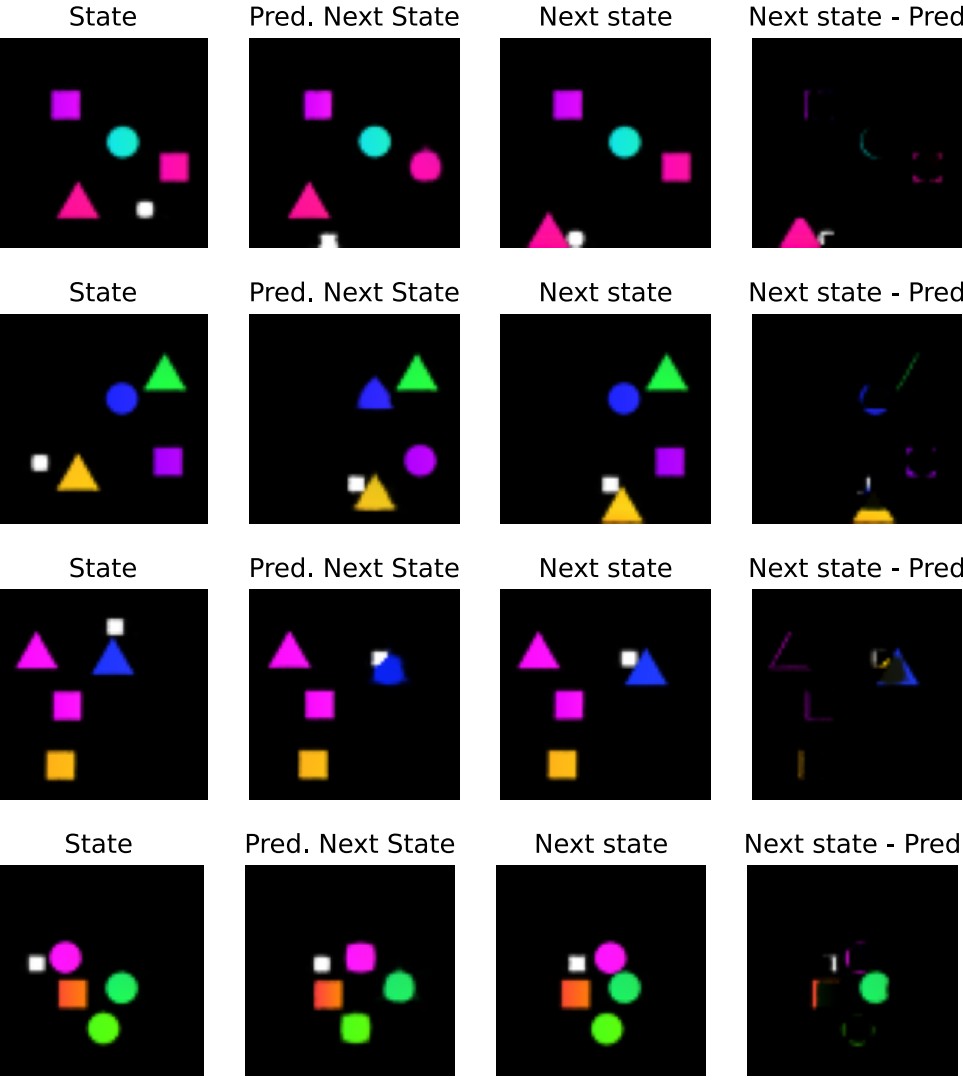

Figure 6: Different visualizations of the behavior of SSWM under the configuration that achieved the results in table 2.

As briefly mentioned in Section 6.3, the metrics used to evaluate the models' performances in latent space can be far from representing the true quality of the transition model. This can be explained by the fact that the transition model is trained to minimize the distance between its predictions and the true next state in latent

space and if it generalizes well in this, it is likely to obtain optimal H@1 and MRR scores during evaluation. However, even if the distances between predictions and target are insignificant for the metrics involved, in compact latent space (where the distance between different states has a small order of magnitude), small perturbations may lead to drastic mispredictions in pixel space as we observe in Figure

### A.2.2  C-SWM

We replicated a similar outcome with the baseline using the same configuration in Table 5 and just 2 feature maps per object.

Table 3: Results obtained by C-SWM using the same configuration in Table 5 made an exception for the number of feature maps per object that in this case is set to 2.

| | 1 Step | | 5 Steps | | 10 Steps | |
|---|---|---|---|---|---|---|
| Test Split | H@1 | MRR | H@1 | MRR | H@1 | MRR |
| test 1 | $100_{\pm 0.0}$ | $100_{\pm 0.0}$ | $100_{\pm 0.0}$ | $100_{\pm 0.0}$ | $100_{\pm 0.0}$ | $100_{\pm 0.0}$ |
| test 2 | $100_{\pm 0.0}$ | $100_{\pm 0.0}$ | $100_{\pm 0.0}$ | $100_{\pm 0.0}$ | $100_{\pm 0.0}$ | $100_{\pm 0.0}$ |
| test 3 | $100_{\pm 0.0}$ | $100_{\pm 0.0}$ | $100_{\pm 0.0}$ | $100_{\pm 0.0}$ | $100_{\pm 0.0}$ | $100_{\pm 0.0}$ |

We observed similar qualitative results with respect to the baseline used in the main experiments. We can indeed notice that the C-SWM encoder did not manage to represent useful information and it fails in assigning each object to a single slot. In this configuration, we observed distances in latent space between a state and its next with an average order of magnitude of $10^{-5}$.

## B  The Importance of Instance Disambiguation

To highlight the importance of disambiguating identical objects, we investigated the performances of C-SWM in Shapes 2D (where C-SWM performs perfectly even after 10 steps). By modifying the dataset to have a unique color (red) for all the shapes (as in figure 1) we forced C-SWM to deal with a situation in which it cannot learn to separate all the shapes in the scene.

Table 4: Results obtained by C-SWM using the same configuration used in Kipf et al. (2019) on the dataset Shapes 2D over a single run.

| | 1 Step | | 5 Steps | | 10 Steps | |
|---|---|---|---|---|---|---|
| Dataset | H@1 | MRR | H@1 | MRR | H@1 | MRR |
| shapes 2d | $64.3_{\pm 0.0}$ | $75.0_{\pm 0.0}$ | $16.9_{\pm 0.0}$ | $30.2_{\pm 0.0}$ | $6.8_{\pm 0.0}$ | $15.6_{\pm 0.0}$ |

The results in Table 4 show that considering couples of similar entities as a single node vector drastically impacts the performances of the transition model. In this way, one object embedding has a single pair of spatial coordinates (in this environment the encoder learns to represent only the position) to describe the location of multiple objects. This makes it difficult for the GNN to generalize as it has to update the vector in such a way that only the object affected by the action is moved. Furthermore, if an action is coupled with a slot containing more objects, there is no information to understand which object is supposed to move. As expected this effect increases in prominence over longer prediction horizons, according to Table 4. It is not possible to show qualitative results in a similar fashion as SSWM as C-SWM was trained without a decoder.

### B.1  Hyperparameters

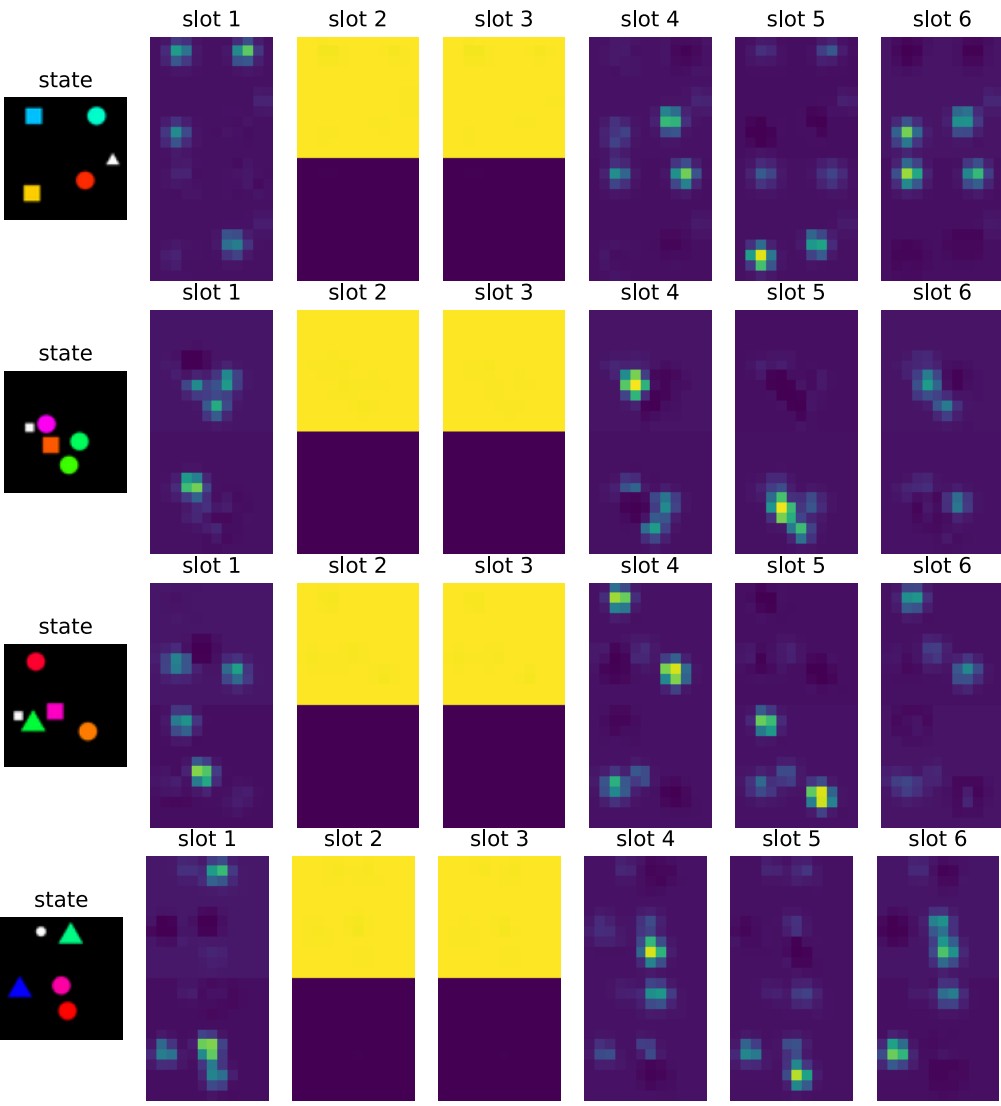

Figure 7: Visualization of the representations produced by C-SWM on different states under the configuration that achieved the results in table 3

Table 5: Hyperparameters for SSWM and C-SWM

|  | Configurations | SA-SWM | C-SWM |
|---|---|---|---|
| Learning | lr | 0.0005 | 0.0005 |
|  | batch size | 512 | 512 |
|  | epochs | 600 | 600 |
|  | num objects (w/bg) | 6 | 6 |
|  | hinge margin | - | 1 |
| CNN Encoder | layer1 activation | - | leaky relu |
|  | layer1 filter size | - | $9 \times 9$ (zero padding) |
|  | layer1 features maps | - | 16 |
|  | layer2 activations | - | sigmoid |
|  | layer2 filter sizes | - | $1 \times 1$ |
|  | layer2 features maps | - | $6 \times 4$ |
| MLP Encoder | hidden dim | - | 512 |
|  | embedding dim | - | 64 |
| Slot Attention Encoder | slots | 6 | - |
|  | iterations | 3 | - |
|  | embedding dim | 64 | - |

