# OpenReview forum: "Slot Structured World Models"
_TMLR — Rejected by TMLR_

### Review · Reviewer_cutr · 2024-08-24

**Summary Of Contributions:**

This paper proposes Slot Structured World Models (SSWM), an object-centric dynamic model that combines slot-based representations from Slot Attention (SA) and a Graph Neural Network (GNN) in the object-centric latent space. Experiments on the Interactive Spriteworld dataset prove that SSWM can outperform a representative baseline, C-SWM, by a sizeable margin consistently.

**Audience:**

Yes

**Broader Impact Concerns:**

No.

**Claims And Evidence:**

No

**Requested Changes:**

Currently, I think the paper needs several major changes to be considered acceptable:
- More proper framing of the proposed method and its novelty. As described above, the combination of GNNs + object-centric representations is not new. What makes this paper unique?
- More experiments on 1) comparing with more baselines, 2) more datasets. Specifically, I would like to see a comparison with SlotFormer [1].
- I would also encourage the authors to experiment with more complicated datasets, such as CLEVRER [2]. The field of object-centric learning has evolved significantly in recent years. I think a paper should not be limited to toy datasets.
- It would also be interesting to see the result of SSWM with a stronger object-centric encoder. Slot Attention does not work well on complex data such as textured objects. I am curious whether SSWM scales to better slot encoders, such as SlotDiffusion [3] and LSD [4].

[1] Wu, Ziyi, et al. "Slotformer: Unsupervised visual dynamics simulation with object-centric models." ICLR. 2023.

[2] Yi, Kexin, et al. "Clevrer: Collision events for video representation and reasoning." ICLR. 2020.

[3] Wu, Ziyi, et al. "Slotdiffusion: Object-centric generative modeling with diffusion models." NeurIPS. 2023.

[4] Jiang, Jindong, et al. "Object-centric slot diffusion." NeurIPS. 2023.

**Strengths And Weaknesses:**

## Strengths

- The paper is well-motivated. Learning an object-centric world model is an important task as it enables more interpretable machine learning systems.
- The proposed method outperforms a baseline C-SWM clearly and consistently. Though I have concern about baseline choice, see Weaknesses.


## Weaknesses

**Limited Novelty**
-  This paper claims to be the first learned dynamic model that "embeds an object-centric encoder and a GNN-based world model". This is not true, as the combination of object-centric features + GNNs has been extensively studied in the literature [1, 2].
- Besides, the paper claims to be the first object-centric dynamic model that can reason object interactions from raw pixels. This is also not true, as [3, 4] can already do so.
- Overall, I find the proposed method lacking technical novelty, especially given the fact that the experimental evaluation is also improper (see below).

**Experiments**
- Baselines: This paper only compares with one baseline C-SWM, which was published in 2020. I believe many important baselines are ignored, such as PARTS [3] and SlotFormer [4]. Although they use a Transformer-based dynamic model, they also belong to the unsupervised object-centric dynamic model family. Without comparing to these recent works, it is hard to assess the proposed method.
- Datasets: This paper only experiments on one dataset. It is a suitable dataset, but I think the authors should at least try one more dataset with physical reasoning. PHYRE [5] would be a good benchmark as it also features object physical interactions and simple visual appearances.

[1] Watters, Nicholas, et al. "Visual interaction networks: Learning a physics simulator from video." NeurIPS. 2017.

[2] Ye, Yufei, et al. "Compositional video prediction." ICCV. 2019.

[3] Zoran, Daniel, et al. "Parts: Unsupervised segmentation with slots, attention and independence maximization." ICCV. 2021.

[4] Wu, Ziyi, et al. "Slotformer: Unsupervised visual dynamics simulation with object-centric models." ICLR. 2023.

[5] Bakhtin, Anton, et al. "Phyre: A new benchmark for physical reasoning." NeurIPS. 2019.

---

### Review · Reviewer_6fbg · 2024-09-09

**Summary Of Contributions:**

This paper investigates objects and inter-object relations in scenes. They propose SSWM, a slot-attention-based model combined with GNNs, to generate object-centric representations for scene objects which can disentangle different yet similar/adjacent objects in the scene. Experiments show the effectiveness of the models.

**Audience:**

No

**Claims And Evidence:**

Yes

**Requested Changes:**

- In the final revision, please ensure the Github link is accessible.
- Please resolve the issues in the figures.
- Please try to argue the importance of the task - why should it still be considered crucial and interesting, given there have been no research updates for five years?

**Strengths And Weaknesses:**

### Strengths
- The idea of applying solt attention may be novel in the current task.
- In the experiments, the evaluations show that the proposed model consistently outperforms baseline C-SWM.

### Weaknesses
- The GitHub link provided is not accessible.
- In Fig.1, the visualization of two rows seems inconsistent. First, the task, i.e., the locations of objects, are different. Also, I wonder (1) why the circles and triangles are shown as squares in the top row, (2) why the colorization mode seems different, and (3) if they are generated with the same code, i.e., in a fair comparison. This hurts the soundness of the paper.
- In Fig.2, there are two "slot attn encoder"'s. Should the latter one be "slot attn decoder"? If not, do they share the same architecture and/or parameters?
- The only dataset used is Interactive Spriteworld. The task images are not photo-realistic and not as challenging as some discussed in previous papers, like "Object-Centric Learning with Slot Attention."
- The only baseline in this paper is C-SWM, which was published in 2019, aka five years ago. Among all the related works, none of them are published in 2024; only 2 of them are published after 2023; only 6 of them are published after 2020. If this is the case, it shows that academia does not consider this task interesting.
- Given the previous two points, I am highly concerned about this paper's satisfaction of the "audience" criterion of the TMLR requirement.

---

> ### Author Response · Authors · 2024-10-10
> **Response to Reviewer 6fbg**
>
> Dear reviewer 6fbg, thank you for your valuable feedback and suggestions.
>
>
> **The GitHub link provided is not accessible.**
> The GitHub link is not accessible only during the review process to ensure the anonymity of the authors. Once the review process is over, we will make the link available. If you are interested in the implementation details, please check the section Supplementary Material, where you can find a zipped and anonymized version of our repository.
>
> **In Fig.1, the visualization of two rows seems inconsistent. First, the task, i.e., the locations of objects, are different. Also, I wonder (1) why the circles and triangles are shown as squares in the top row, (2) why the colorization mode seems different, and (3) if they are generated with the same code, i.e., in a fair comparison. This hurts the soundness of the paper.**
> Regarding the different environments, the caption of the figure states indeed that the C-SWM object extractor was evaluated on Shapes 2D while the SSWM encoder (Slot Attention) was on Interactive Spriteworld (IS). The reason behind this choice is that C-SWM did not learn any useful features in IS, and then it was not informative to make the comparison in the same environment. Due to the easier dynamics of shapes 2D C-SWM could learn good representation and therefore we could easily show the disambiguation problems when dealing with identical objects. On the other hand, we did not find it necessary to train slot attention (for this purpose) on shapes 2D as this model is already well-tested in the original paper and subsequent extensions. The purpose of the figure is to show that in a situation where there are multiple objects with identical appearances, one method manages to separate them into different slots and one cannot (as C-SWM relies only on a CNN, identical objects will be detected by the same feature map). We updated the caption to briefly mention this.
> Regarding the difference in colors, we converted the C-SWM feature maps into grayscale to improve consistency. This does not change the meaning behind the figure, only the color mapping.
>
> **In Fig.2, there are two "slot attn encoder"'s. Should the latter one be "slot attn decoder"? If not, do they share the same architecture and/or parameters?**
> No, the two "slot attn encoder"s are the same encoder (same parameters). In the left part of the figure the encoder is used to obtain the object embeddings from the current state, which are then fed to the iterative GNN, while on the right side, it produces the object embeddings from the state subsequent to the current one (these are used as targets in the L2 latent loss). This is the same strategy used in the C-SWM paper. We updated the caption to highlight the fact that the two encoders in the figure are identical.
> The only dataset used is Interactive Spriteworld. The task images are not photo-realistic and not as challenging as some discussed in previous papers, like "Object-Centric Learning with Slot Attention."
> We completely agree on this point, unfortunately, due to limited time and computational resources, evaluating the method on more complex environments over multiple independent runs was out of our possibilities.
>
> **The only baseline in this paper is C-SWM, which was published in 2019, aka five years ago. Among all the related works, none of them are published in 2024; only 2 of them are published after 2023; only 6 of them are published after 2020. If this is the case, it shows that academia does not consider this task interesting.**
> The paper was written in 2023 and therefore it is possible that some recent valuable works did not get our attention. Nonetheless, we do not agree on the fact that a not very active research area should be considered less important.

---

### Review · Reviewer_pX1Q · 2024-09-27

**Summary Of Contributions:**

The paper proposes to combine Slot Attention with Structured World Models for the task of predicting the effects of object interactions from visual representations. The evaluation is performed in the Spriteworld environment of moving 2D colored shapes, extended with a physical interactions where the objects can push one another. The authors compare with a single baseline, which is a similar architecture that uses a different encoder. The results show that the proposed method consistently outperforms the baseline in quantitative metrics, and qualitatively that it can capture the physical interactions between different objects.

**Audience:**

Yes

**Claims And Evidence:**

No

**Requested Changes:**

I would ask for two additional experiments:
1. A qualitative evaluation demonstrating how a failure of C-SWM leads to bad predictions, while SSWM succeeds.
2. An experiment showing that SSWM produces good results when tested with higher numbers of objects than it sees during training.

**Strengths And Weaknesses:**

The paper is clear and the contribution is easy to understand, and the proposed method shows a clear improvement over the baseline. I am not particularly familiar with this area, so I can not evaluate whether the baseline is appropriate and whether the paper is correctly positioned within the existing literature. I think the main weaknesses have to do with how well the experiments support the claims of the paper.

The first claim is that the encoder used in C-SWM can not distinguish different objects with similar appearance. While some evidence of that is presented in Figure 5, I would really like to see the downstream effect of this problem in something similar to Figure 4.

The second claim is that SSWM is not limited to a fixed number of slots, while C-SWM is. However, it seems that in practice the authors fix the number of slots for convenience anyway. I think in practical terms, having an architecture with an unbounded number of slots only matters if it can be shown to generalize to higher number of objects than it ever sees at training time, and I don't see such an experiment in the paper.

---

> ### Author Response · Authors · 2024-10-10
> **Response to Reviewer pX1Q**
>
> Dear reviewer pX1Q, thank you for your valuable feedback and critiques. We completely agree with your comment and worked to satisfy at least one of your requests.
>
> We added a section in the appendix to show how the instance disambiguation problem affects the downstream task. In this section, we provide quantitative evidence of this effect on Shapes 2D (on which C-SWM has perfect scores) modified such that each scene in the dataset has two couples of identical objects. As explained in the section we could not provide a qualitative analysis similar to the one provided for SSWM, as C-SWM was trained without a decoder.
>
> Regarding your second request, unfortunately, due to the limited time and computational resources in our availability, we did not manage to perform the experiments on the variable number of objects. We understand the relevance of this point and apologize for not fulfilling it.

---

### Decision · Action_Editor_bD4M · 2024-12-09

**Recommendation:** Reject

**Comment:**

While the paper presents a very interesting idea, the paper claims that this is the first work that can isolate individual objects and reason about their (action-conditional) interactions from raw pixel input. Reviewers note that this claim does not match the evidence provided, in light of existing work. Reviewers also note that the paper would be strengthened by more challenging environments, eg CLEVRER. Responses by the authors did not manage to sufficiently address these concerns, so this paper needs one more significant iteration of work before it is ready for publication.

**Audience:**

This paper will be interesting to an ML and vision audience, so definitely relevant to TMLR.

**Claims And Evidence:**

The claims in this paper that the proposed Slot Structured World Model can make relatively accurate predictions, even on similar objects, match the experimental evidence. The additional claim that this is the first model to do so, is not matched by evidence, as reviewers pointed to a number of papers with similar models in the last couple of years, which limits the novelty of this paper.